# Diagnostic Performance of a Fecal Immunochemical Test-Based Colorectal Cancer Screening Program According to Ambient Temperature and Humidity

**DOI:** 10.3390/cancers14051153

**Published:** 2022-02-23

**Authors:** Gemma Ibáñez-Sanz, Núria Milà, Núria Vives, Carmen Vidal, Gemma Binefa, Judith Rocamora, Carmen Atencia, Víctor Moreno, Rebeca Sanz-Pamplona, Montse Garcia

**Affiliations:** 1Oncology Data Analytics Programme, Catalan Institute of Oncology, Hospitalet de Llobregat, 08907 Barcelona, Spain; gemma.ibanez@bellvitgehospital.cat (G.I.-S.); jrocamora.caballero@iconcologia.net (J.R.); catencia@iconcologia.net (C.A.); v.moreno@iconcologia.net (V.M.); rebecasanz@iconcologia.net (R.S.-P.); 2Gastroenterology Department, Bellvitge University Hospital, Hospitalet de Llobregat, 08907 Barcelona, Spain; 3Colorectal Cancer Research Group, ONCOBELL Programme, Institut d’Investigació Biomèdica de Bellvitge (IDIBELL), Hospitalet de Llobregat, 08907 Barcelona, Spain; 4CIBER Epidemiology and Public Health (CIBERESP), 28029 Madrid, Spain; nmila@iconcologia.net (N.M.); gbinefa@iconcologia.net (G.B.); 5Cancer Screening Unit, Prevention and Control Programme, Catalan Institute of Oncology, Hospitalet de Llobregat, 08907 Barcelona, Spain; nvives@iconcologia.net (N.V.); cvidal@iconcologia.net (C.V.); 6Early Detection of Cancer Research Group, EPIBELL Programme, Institut d’Investigació Biomèdica de Bellvitge (IDIBELL), Hospitalet de Llobregat, 08907 Barcelona, Spain; 7Department of Clinical Sciences, Faculty of Medicine and Health Sciences, University of Barcelona, 08907 Barcelona, Spain

**Keywords:** fecal immunochemical test, screening program, interval colorectal cancer, temperature, humidity

## Abstract

**Simple Summary:**

Hemoglobin degradation can be affected by ambient temperature and humidity. How this modifies the advanced neoplasia detection rate and interval cancer rate remains understudied. We conducted a retrospective study and analyzed the impact of ambient temperature and humidity on the fecal immunochemical test (FIT) positivity rate, detection rate for advanced neoplasia, and interval colorectal cancer (CRC). The results of our study indicated that at >24 °C, the positivity rate was lower, whereas the detection rate of the FIT for advanced neoplasia and the interval cancer detection rate were not affected, probably because we have adopted measures to minimize the impact of ambient temperature on FIT sensitivity. Humidity did not affect FIT sensitivity. The results emphasize the importance of organizational efforts on the procedures along the screening process (such as the cold chain) to minimize the effect of seasonal variations in temperature on the positivity rate.

**Abstract:**

Exposure of the fecal immunochemical test (FIT) to different ambient temperatures and humidity is unavoidable in population-based screening programs in Southern European countries, and it could lead to a decrease in target colorectal lesions. The objective was to evaluate the effect of ambient temperature and humidity on the FIT sensitivity in a population-based screening program for colorectal cancer (CRC) using an ecological design. The retrospective cohort included individuals aged 50–69 years who participated in CRC screening (Barcelona) from 2010–2015, and were followed until 2017 to identify interval CRCs. The positivity rate, and detection rates for advanced polyps and CRC were compared according to ambient temperature, humidity, and quarters of the year. A positive FIT was defined as the detection of ≥20 μg Hb/g in feces. The monthly ambient temperature and humidity were recorded on the day that the FIT was performed. In total, 92,273 FIT results from 53,860 participants were analyzed. The FIT positivity rate was lower at >24 °C than at ≤24 °C (*p* = 0.005) but was not affected by humidity. The temperature’s impact on positivity did not lead to a decrease in the FIT detection rate for advanced neoplasia or the interval cancer detection rate in a program where the samples were refrigerated until the analysis and screening invitations were discontinued in July and August.

## 1. Introduction

Colorectal cancer (CRC) screening based on fecal occult blood followed by a diagnostic colonoscopy reduces CRC mortality [1]. The fecal immunochemical test (FIT) is the preferred screening test for CRC in most organized screening programs [2,3]. The interval cancer rate is strongly correlated with the sensitivity of the FIT and reflects the quality of the screening program. In a population-screening program, the FIT has a sensitivity of 29% and a specificity of 97% for detecting advanced adenoma, and a sensitivity of 86% and a specificity of 85% for detecting CRC [4,5]. Studies of FIT stability [6,7,8,9,10] and the FIT manufacturers’ specifications suggest shorter intervals between collection and testing as continued exposure to ambient temperature decreases the test performance of the FIT. Several studies, mainly with a cross-sectional design, have examined the FIT’s performance when samples are returned during warm months with inconsistent results. A few of them suggest that the positivity rate of the FIT is reduced with high ambient temperatures [11,12,13,14,15], but some other investigations have reported no significant variations in positivity rates according to seasonal temperatures [16,17,18,19]. Moreover, Adam et al. [20] suggested that humidity could also be important in maintaining the performance of FIT in an experimental study. Only Park et al. [17] has analyzed the effect of temperature and humidity in a CRC screening setting. They reported that high temperature and high humidity decreased FIT’s positivity rate.

The existing literature on the impact of ambient temperature on FIT and advanced neoplasia detection in a population-based screening program is scarce [13,14,15,21]. Only two studies [13,14] have evaluated the risk of interval cancer, suggesting, though with caution, that interval cancers were more frequently detected in the summer and autumn seasons. In fact, the US Multi-Society Task Force [22] states insufficient evidence to recommend against distributing or mailing FITs when ambient temperatures are above a certain level.

This study aimed to evaluate the effect of ambient temperature and humidity, both independently and in combination, on the FIT’s sensitivity using a population-based screening cohort. The positivity rate and FIT’s sensitivity for advanced polyps and CRC were assessed according to ambient humidity and temperature.

## 2. Materials and Methods

### 2.1. Target Population

A biennial screening program for CRC, which is free of charge, is managed by the Catalan Institute of Oncology. We used the FIT with a cut-off of ≥20 μg Hb/g feces (100 ng/mL) (OC-Sensor, Eiken Chemical Co., Tokyo, Japan) as the screening test. The target population of the current study (*n* = approximately 495,000) includes men and women aged 50–69 years from the Northern and Southern metropolitan areas of Barcelona (Catalonia, Spain). The exclusion criteria for the screening program were gastrointestinal symptoms, advanced polyps’ history, hereditary CRC syndromes, familial or personal history of CRC, inflammatory bowel disease, colonoscopy in the previous five years or a FIT within the last two years, terminal illness, and severely disabling conditions. The description of our screening program and its quality indicators has been described previously [23,24].

### 2.2. Screening Invitation Process

All eligible subjects received an invitation letter to collect a FIT kit at any nearby community pharmacy. We sent the invitations according to the primary healthcare areas (territorial divisions through which primary health care services are organized). Individuals in a given geographic area were assigned to a primary care team and an endoscopic unit for the diagnostic colonoscopy. Once the sample was collected, subjects were instructed to store the sample in the refrigerator and take it to a nearby pharmacy as soon as possible. The sample was exposed to ambient temperatures during transport from home to pharmacy. Subsequently, it was transported refrigerated to a central testing center by a courier. The distribution was daily or at least three times a week. Samples were rejected if there was a delay of ≥12 days from the time of collection to the laboratory, and patients were sent a new kit to collect a new sample. Among participants who completed the FIT, the median time of the return of the kit was three days. Appendix A shows the conservation temperature, maximum days for each step of the process, and if the temperature was controlled. During the summer holidays (July and August), the program usually does not send invitations. However, the participation rate is similar in the third quarter (July–September) and the first quarter (January–March).

### 2.3. Study Population

For the study, we selected individuals with a conclusive FIT result from October 2009 to December 2015. The minimum follow-up was 24 months. Endoscopic findings were classified according to the European guidelines for quality assurance in CRC [25]. Low-risk lesions were defined as one or two tubular adenomas measuring <10 mm with low-grade dysplasia; intermediate-risk lesions were defined as three or four tubular adenomas measuring <10 mm with low-grade dysplasia, or as one to four adenomas measuring 10–19 mm with low-grade dysplasia or at least one with tubulovillous/villous or carcinoma in situ or with high-grade dysplasia; high-risk lesions were defined as ≥5 adenomas or ≥1 adenoma measuring ≥20 mm. Subjects were classified according to their most advanced lesion. The CRCs were divided into (1) screen-detected cancer: cancer detected after a positive FIT, and (2) interval cancer: cancer diagnosed after a negative FIT result and before the next screening invitation (≤24 months). We defined a post-colonoscopy CRC as a CRC diagnosed after a diagnostic colonoscopy in which no CRC was detected before the next recommended exam date. As per [25,26,27], positive FITs without subsequent colonoscopy or with a low-quality colonoscopy (incomplete or inadequate bowel preparation), post-colonoscopy CRCs, CRCs diagnosed before the first participant’s invitation, and those cancers diagnosed more than 24 months after screening FIT analysis were not included in the analyses.

Our CRC screening program follows the Public Health laws and the Organic Law on Data Protection. All procedures performed in the study involving data from human participants were in accordance with the ethical standards of the institutional research committee and with the 1964 Helsinki Declaration and its later amendments or comparable ethical standards. No informed consent was requested since this study was based on anonymized data that is routinely collected. The study protocol was approved by the Ethics Committee for Clinical Research (PR234/16).

### 2.4. Collection of Weather Information

Data regarding the monthly temperature and relative humidity (average, minimum and maximum) during the study period were registered by the Automatic Weather Stations Network of Catalonia (METEOCAT and L’Hospitalet weather station). Averages of the daily maximum temperatures and maximum relative humidity were divided into ≤24 °C and >24 °C for temperature and ≤89% and >89% for humidity.

### 2.5. Data Analysis

The following variables were included in the analysis: sex, age, deprivation index, date the FIT was performed, the quantitative result of the FIT, screening episode (first invitation or successive), colonoscopy findings, date of CRC diagnosis, and weather parameters (temperature, humidity, and quarter of the year). The deprivation index was calculated for the primary healthcare areas of Catalonia [28] with a scale from 0 (least deprived) to 100 (most deprived). This index uses aggregated income, health, disability, employment, and education indicators. Our CRC screening program database provided the following data: the FIT result, the number of individual participations, the date of the FIT, and colonoscopy findings. Then, this data was linked with information on procedures and diagnoses of public hospitals of Catalonia [29] (the minimum basic set of hospital discharge data (CMBD-AH). An exhaustive chart review of all the CRC cases was performed to avoid misclassifying a polyp into the CRC category. Subjects were followed until February 2017 to identify whether they were diagnosed after a FIT with a CRC (International Classification of Diseases 10th Revision (ICD-10): C18, C19, and C20: colon or rectum). Anal and appendix cancers were excluded.

We examined the positivity rate, neoplasia detection rate, and interval cancer rate related to temperature and humidity. As the hemoglobin concentration was not normally distributed and logarithmic transformation did not achieve data normalization, bootstrapping techniques to calculate the mean and confidence intervals were used. A non-parametric test (Kruskal–Wallis rank test for equality) was used to evaluate the difference across quarters of the year and temperature levels. Multiple logistic regression was used to assess the influence of FIT positivity according to weather parameters, adjusting for sex, age, screening episode (first or successive screening), and deprivation index. Multiple logistic regression was also used to evaluate the probability of an interval CRC versus a screening-detected CRC according to temperature and to adjust for sex, age at diagnostic, screening episode, and deprivation index. We performed a sensitivity analysis including temperature and humidity as continuous variables, obtaining similar results. We presented the results as adjusted odds ratios (ORs) and 95% confidence intervals (CI) using logistic regression. Statistical analysis was carried out using R statistical software (R Foundation for Statistical Computing, Vienna, Austria).

## 3. Results

A total of 53,860 participants of the CRC screening program had at least one conclusive FIT result (positive or negative) (*n* = 92,273). Figure 1 shows a flowchart of the population examined, from when they were invited to perform the FIT to when they were diagnosed with a CRC. The FIT resulted in 5048 (5.5%) positive tests (4991 individuals), but in 530 (10.5%) of them, the colonoscopy was not performed, either because of refusal (*n* = 353) or it was not indicated for medical reasons (*n* = 177). These subjects were excluded from the analysis.

Baseline characteristics of participants according to the FIT result are shown in Table 1. Screen-detected CRCs, high-risk lesions, and intermediate-risk lesions as most advanced findings were found in 211 (4.6%), 822 (18.2%), and 1118 (24.7%) participants, respectively. In 762 (16.9%) and 1605 (35.5%) participants, low-risk lesions and no preneoplastic lesions were found, respectively.

Appendix A shows the main characteristics of the participants according to temperature and humidity. The median average temperature throughout the year was 14 °C, and relative humidity was 65%. The average high ambient temperature was 19 °C (25th and 75th percentiles 17, 21) in January–March; 27 °C (24, 29) in April–June; 31 °C (30, 33) in July–September; and 21 °C (18, 20) in October–December. The average high ambient humidity was 82% (25th and 75th percentiles 78, 85) in January–March; 84% (83, 85) in April–June; 84% (82, 86) in July–September; and 86% (81, 89) in October–December.

A high temperature was defined as >24 °C and high humidity as >89%. Figure 2a,b shows the monthly fluctuations in positivity for the FIT over the year and the monthly temperature and relative humidity during the study period, respectively. The average temperature was highest in August, July, June, and September. Relative humidity remained remarkably stable during the year. The lowest positivity rates were in June, July, and August, which are the months with the higher temperatures. However, September had a similar temperature, and the FIT positivity was higher. The positivity rate of the FIT with a cut-off of ≥20 μg Hb/g feces was significantly lower in temperatures > 24 °C (5.2%) than in ≤24 °C (5.6%) (*p* = 0.005).

Table 2 shows the mean hemoglobin concentration according to the seasons. The lowest concentration was recorded in the third quarter (July–September) (mean 27.2 ng/mL (95% CI 24.6 to 29.7)) and the highest in the first quarter (January–March) (mean 30.5 ng/mL (95% CI 29.1 to 32.0)).

The results of the multivariate analysis (Table 3) for the positivity rate showed that the covariates associated with a higher positivity rate were male sex, advanced age, a lower socioeconomic status, and temperature > 24 °C (OR adjusted by sex, age, deprivation index, and successive screening: 0.88; 95% CI, 0.83–0.94). The multivariable analysis did not include the interaction between high ambient temperature and humidity (*p* = 0.83). Ambient humidity did not affect the positivity rate of FIT in the univariate or multivariable analysis. The results of the logistic regression of the probability of positive screening tests by quarter of the year (Appendix A) show that there was a 15% lower probability of the FIT being positive in July–September than in January–March.

We also assessed the impact of ambient temperature on the two-year FIT sensitivity. The detection rate of the FIT for advanced polyps (including screen-detected CRCs, high-risk lesions, and intermediate-risk lesions) decreased but not significantly with temperature (≤24 °C detection rate of 23.65 (95% CI 22.50–24.87) vs. >24 °C detection rate of 22.54 (95% CI: 20.85–24.35). No differences were found for humidity (≤89% detection rate of 23.25 (95% CI 22.23–24.31) vs. >89% detection rate of 23.77 (95% CI 21.03–26.86)). Finally, we observed that the probability of detecting advanced neoplasia was similar by temperature (Table 4) or different quarters of the year (Appendix A).

As Figure 1 shows, there were 51 interval CRCs and 211 screen-detected CRCs. We excluded the following from the analyses: 19 CRCs after a FIT positive result followed by a colonoscopy without CRC (post-colonoscopy CRC), 10 CRCs diagnosed in FIT positive participants who did not undergo a colonoscopy; and 64 diagnosed more than 24 months after screening FIT analysis (individuals with a negative FIT result who did not participate in successive screening). FIT sensitivity for CRC during this period using the interval cancer proportion method was 19.5%. Interval cancers were detected in ≤12 months of the FIT being performed in 33% of the cases (*n* = 17), and 67% (*n* = 34) of them were diagnosed after 12 months. Compared with the screen-detected group, there were no differences in the quarter of the year when the FIT was performed and the detection of interval CRCs (*p* = 0.11). Moreover, no differences were found when comparing screen-detected and interval CRCs according to ambient temperature (OR adjusted by sex, age, deprivation index, and successive screening: 1.72; 95% CI, 0.83–3.57).

## 4. Discussion

In this study, we analyzed the association between ambient temperatures and the performance of FIT in a CRC screening population. As already mentioned in the introduction, much of the previous research has been restricted to assessing the effect of ambient temperature on the FIT positivity rate. The findings of this investigation complement those of earlier studies as we have also analyzed the impact of ambient temperature and humidity on the two-year sensitivity of the FIT. The findings showed that the positivity rate of the FIT with a cut-off of ≥20 μg Hb/g feces was slightly lower when the ambient temperature was >24 °C, which is consistent with some previous reports [11,12,13,14,15]. However, monthly variations in temperature or humidity when the FIT was performed did not modify the detection rate for advanced neoplasia (CRCs, high-risk lesions, and intermediate-risk lesions). Additionally, the interval cancer detection rate was similar regardless of temperature or humidity. It is important to highlight that our screening program did not send invitations during the warmer months (summer holidays), so the July and August participation rate was very low. Moreover, we instructed participants to return the sample to the pharmacy as soon as possible and to refrigerate it. This might influence how the impact of the decrease in positivity is not causing a decrease in the detection rates.

Studies focusing on the impact of ambient temperature on FIT and advanced neoplasia detection are scarce. An experimental study [30] that compared the detection rate of advanced neoplasia between low (<10 °C) and high (≥25.0 °C) temperature groups among FIT participants concluded that high ambient temperature was not a risk factor for either a positive FIT result or the detection of advanced neoplasia. In population-based screening programs, only four studies have analyzed the impact of temperature on advanced neoplasia detection rates [13,14,15,21] and two have analyzed the impact on interval cancer rates [13,21]. The study of Cha et al. [13], where subjects were instructed to return the FIT sample rapidly and store it in a refrigerator, analyzed around five million FITs of different brands and several cut-off points. Moreover, they only analyzed the data according to seasons but not by temperature. They reported that cancer detection rates were not influenced by season. When quantitative FITs were analyzed, interval cancers were more frequently detected in the summer and autumn seasons than in the winter. Nevertheless, the impact on interval cancer rates is difficult to interpret as the data included different cut-offs and FIT brands. Doubeni et al. [14] reported that the FIT’s sensitivity for CRC was significantly lower in June/July (75%) than in December/January (79%), but participants were not instructed to refrigerate the collected samples. In the Doubeni et al. [14] and the Cha et al. [13] studies, the sensitivity of FIT could be overestimated as they defined CRC diagnosis as within 12 months of the test date and not 24 months. In Niedermaier et al. [15], participants were asked to send the samples by regular mail; they examined sensitivity and specificity for advanced neoplasia at five different cut-offs and according to the maximum temperature while returning the FIT and according to the interval time to return the sample. Hemoglobin concentrations were lower in individuals with advanced neoplasia when FIT samples were exposed to ≥25 °C, compared to <10 °C. Although they observed that the differences in sensitivity stratified by the temperature at FIT higher positivity thresholds were apparently stronger, they were not significant. Finally, Bretagne et al. [21] suggested that the spring/summer was a risk factor for interval cancers but warned that this result must be interpreted cautiously as the confidence intervals were very wide (*n* = 209 interval cancers), and the data were from two rounds of guaiac fecal occult blood test and one round of FIT (cut-off of ≥30 μg of hemoglobin per gram of feces). Therefore, comparability with previous studies is difficult due to differences in the temperature and storage time the samples are subjected to and the country’s weather conditions.

Ambient humidity did not affect FIT’s positivity rate in our screening population. This contrasts with the results of a cohort in Korea [17]. They found that high temperature and humidity were associated with a low positivity rate of FIT, but neither high temperature alone nor high humidity alone affected the positivity rate. Since we did not detect differences in positivity according to humidity in the univariate analysis, we did not include the interaction between humidity and temperature in the multivariate analysis. No noticeable differences in humidity in our study could be explained by the fact that the humidity in Spain is high but stable (around 65%) during the year, while in Korea it varies from 60% to 90% depending on the month of the year. Park et al. [17] could not perform a specific analysis with the cancer detection rate as they only included 567 subjects with advanced adenoma and 27 with CRC.

Studies of FIT stability [6,7,8,9,10] have showed that exposure to ambient temperature decreased the FIT’s test performance. Some previous studies in population-based screening programs suggest, though with caution, that interval cancers rates may increase with higher temperatures. We have observed that the decrease in positivity did not impact either the detection rate of the FIT for advanced polyps or the interval cancer detection rate. Although we cannot measure the actual effect of the cold chain strategy in our findings, it can be assumed that maintaining the cold chain and reducing the period between collection and measurement during FIT performance minimized the impact of temperature on FIT sensitivity. For this reason, one of the issues that emerge from our and previous findings is the need to implement effective measures to overcome the suboptimal performance of FIT during high ambient temperatures. This is of paramount importance given that it is predicted that temperature means and extremes are projected to be higher (1.5–2 °C) in the coming years [31]. Screening programs, especially in those where the FITs are returned by mail, should implement circuits so that temperature does not impact the detection of advanced neoplasia. More effective measures could be used to optimize FIT performance, as Grazzini et al. [11] proposed, such as (a) decreasing the cut-off levels of FIT during warmer months; (b) reducing the interval of 2 years between screening episodes for those tested during warmer months; and (c) modifying the period of invitation so that a participant with a FIT performed in a warm month for the first personal screening round would be invited during the cold months for the next screening round. Of all the measures that could be adopted, we believe that the most feasible to implement would be to instruct subjects to guarantee the cold chain after sample collection, decrease the number of FITs performed during warm months and decrease the cut-off during warm periods. Other measures could increase program complexity.

The generalizability of these results is subject to certain limitations. One weakness of this study that could have affected the temperature measurements is that we only had monthly temperature data and not daily. We did not have data on the time of fecal sampling to adjust for sample delay. Another concern is that temperature fluctuations in the first to the second stage of the process (sample storage at home at 2–8 °C and delivery to the pharmacy at ambient temperature) could also impact the screening results. Still, we could not investigate this pre-analytical factor in a population-based screening program. Experimental studies should evaluate this effect. On the other hand, we did not have detailed information on humidity (air conditioning or dehumidifier devices) which could also impact the results. Program invitations are distributed according to geographical areas, but these differences have been minimized by adjusting for an aggregated deprivation index. Finally, the number of CRCs cases was relatively small, although we included a large number of subjects, and the proportion of interval cancer was consistent with previous literature [32].

## 5. Conclusions

The positivity rate of the FIT decreased with high ambient temperature. Still, it did not affect the detection rate of the FIT for advanced polyps or the interval cancer rate, probably because we have adopted measures to minimize the impact of ambient temperature on positivity rates. Humidity was not associated with a lower positivity rate. However, further prospective studies are needed to evaluate the effect of temperature specifically on the interval cancer detection rates among biennale FIT-based programs with different screening procedures.

## Figures and Tables

**Figure 1 cancers-14-01153-f001:**
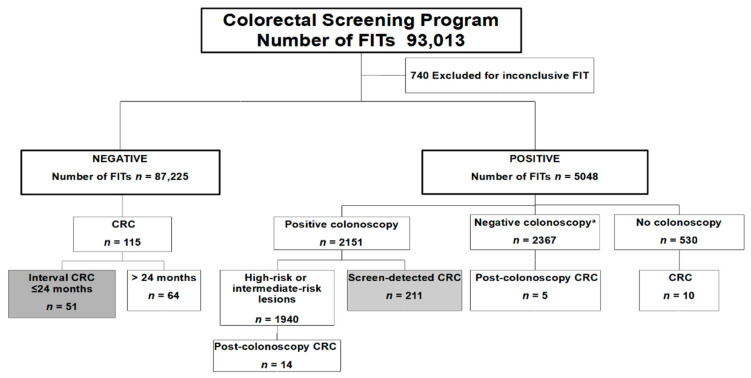
Flow diagram of the participants included in this analysis. CRC: colorectal cancer; FIT: Fecal immunochemical test. ^a^ Negative colonoscopy was defined as having low-risk lesions or no preneoplastic lesions.

**Figure 2 cancers-14-01153-f002:**
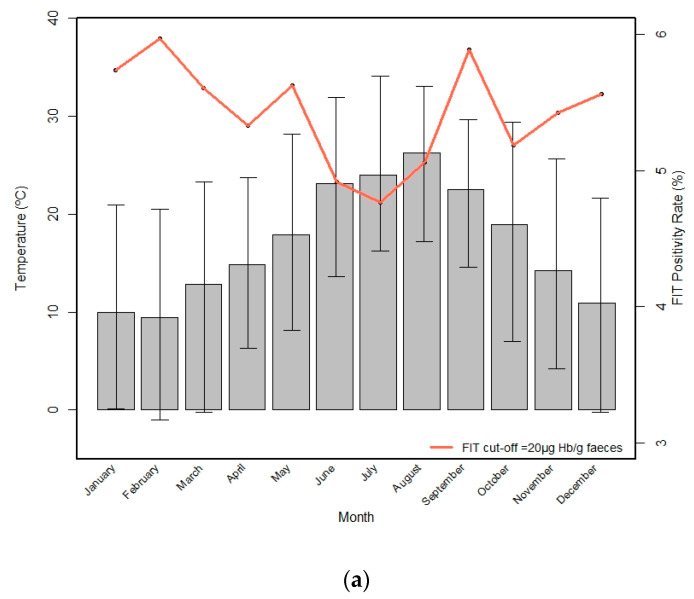
Monthly fluctuations in positivity for the FIT over the year (red line) and the monthly temperature (**a**) and humidity (**b**) during the study period, respectively. The extremes in the confidence intervals represent the minimum and maximum temperature of each month.

**Table 1 cancers-14-01153-t001:** Baseline characteristics of participants according to the FIT result.

Variation	Negative	Positive
	*N* = 87,225	*N* = 5048
Sex		
Female	48,789 (55.9%)	2185 (43.3%)
Male	38,436 (44.1%)	2863 (56.7%)
Age (years)		
50–59	42,485 (48.7%)	2095 (41.5%)
60–69	44,740 (51.3%)	2953 (58.5%)
Screening episode		
Initial	50,554 (58.0%)	3306 (65.5%)
Successive	36,671 (42.0%)	1742 (34.5%)
Socioeconomic Score		
0–39 (least deprived)	6671 (7.65%)	378 (7.49%)
39–51	33,665 (38.6%)	1913 (37.9%)
52–100 (most deprived)	46,889 (53.8%)	2757 (54.6%)
Quarter year of FIT performance		
1st quarter (January–March)	27,278 (31.3%)	1675 (33.2%)
2nd quarter (April–June)	18,091 (20.7%)	1007 (19.9%)
3rd quarter (July–September)	8376 (9.60%)	449 (8.89%)
4th quarter (October–December)	33,480 (38.4%)	1917 (38.0%)
Maximum ambient temperature		
≤24 °C	60,375 (69.2%)	3589 (71.1%)
>24 °C	26,850 (30.8%)	1459 (28.9%)
Maximum ambient humidity		
≤89%	76,933 (88.2%)	4446 (88.1%)
>89%	10,292 (11.8%)	602 (11.9%)

FIT: fecal immunochemical test.

**Table 2 cancers-14-01153-t002:** Variation in hemoglobin (ng/mL) concentration according to the quarter of the year when the FIT was performed.

Quarter Year of FIT Performance	Mean (95% CI) Hb	Number of FIT Results	Min Hb	Max Hb	p25	p50	p75
1st quarter (January–March)	30.5	28,953	0	1000	0	0	10
2nd quarter (April–June)	28.9	19,098	0	1000	0	0	5
3rd quarter (July–September)	27.2	8825	0	1000	0	0	4
4th quarter (October–December)	28.8	35,397	0	1000	0	0	7

Hb: Hemoglobin (ng/mL); FIT: fecal immunochemical test; p25, p50, p75 are 25th, 50th and 75th percentiles, respectively. Kruskal–Wallis: *p* < 0.001.

**Table 3 cancers-14-01153-t003:** Logistic regression of the probability of positive screening tests by temperature and humidity (95% CI).

Variation	Number of FITs	OR ^2^	95% CI	*p*-Value
	*N* = 5048			
Sex				
Female	2185 (43.3%)	1.00		<0.001
Male	2863 (56.7%)	1.67	1.58–1.77	
Age ^1^ (years)	60.7 (5.8)	1.03	1.03–1.04	<0.001
Screening episode				
Initial	3306 (65.5%)	1.00		<0.001
Successive	1742 (34.5%)	0.68	0.64–0.72	
Socioeconomic score ^1^	53.5 (10.9)	1.01	1.00–1.01	<0.001
Maximum ambient temperature				
≤24 °C	3589 (71.1%)	1.00		<0.001
>24	1459 (28.9%)	0.88	0.83–0.94	
Maximum ambient humidity				
≤89%	4446 (88.1%)	1.00		0.06
>89%	602 (11.9%)	1.09	1.00–1.19	

^1^ Continuous variable; ^2^ All variables shown are included in the multivariate analysis. FIT: fecal immunochemical test.

**Table 4 cancers-14-01153-t004:** Logistic regression of advanced neoplasia (screen-detected CRCs, high-risk and intermediate-risk lesions).

Variation	No Advanced Neoplasia	Advanced Neoplasia	OR ^2^	95% CI	*p*-Value
	*N* = 2367	*N* = 2151			
Sex					
Female	1278 (54.0%)	678 (31.5%)	1.00		
Male	1089 (46.0%)	1473 (68.5%)	2.53	2.24–2.86	<0.001
Age ^1^ (mean (SD)) (years)	60.6 (5.8)	60.8 (5.8)	1.01	1.00–1.02	0.01
Screening episode					
Initial	1422 (60.1%)	1515 (70.4%)	1.00		
Successive	945 (39.9%)	636 (29.6%)	0.63	0.56–0.72	<0.001
Socioeconomic score ^1^ (mean (SD))	53.3 (10.8)	53.5 (11.1)	1.00	1.00–1.01	0.54
Maximum ambient temperature					
≤24 °C	1706 (72.1%)	1513 (70.3%)	1.00		
>24 °C	661 (27.9%)	638 (29.7%)	1.08	0.94–1.25	0.26

^1^ Continuous variable; ^2^ All variables shown are included in the multivariate analysis. FIT: fecal immunochemical test

## Data Availability

The data presented in this study are available on request from the corresponding author.

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
