# Peer review of "Diagnostic Performance of a Fecal Immunochemical Test-Based Colorectal Cancer Screening Program According to Ambient Temperature and Humidity"

_cancers, 2022, doi:10.3390/cancers14051153_

Round 1

Reviewer 1 Report

Dear authors,

It is a pleasure for me to Review your manuscript. Only some details that I did not understand very well.

Figure 1. Negative FITs = 87 225/ CRC= 77/ IC <= 24 months 51 + > 24 months 64 ???? Could you review the results paragraph as well?

Regards

Author Response

Reviewer #1:

It is a pleasure for me to Review your manuscript. Only some details that I did not understand very well. Figure 1. Negative FITs = 87 225/ CRC= 77/ IC <= 24 months 51 + > 24 months 64 ???? Could you review the results paragraph as well?

Thanks for noticing it. The mistake in Figure 1 has been corrected. The sum of CRCs among FIT negative participants is 115. The results paragraph does not have any mistake as the sum is not mentioned.

Reviewer 2 Report

This is a well written paper focused on the Fecal immunochemical based test (FIT) as screening tool for colorectal cancer (CRC). The Authors investigated if, in a large number of cases, temperature and humidity could affect the accuracy of the test. The final question the Authors ask themselfs is if the screening organization has to spend any effort to mantain the fecal sample in a range of temperature and humidity in order not to decrease the screening program efficacy. The results of the work of Ibanez-Sanz et al. are that during summer there is a lower rate of positive FIT but not a lower adenomas and interval carcinomas detection rate. 

The methods are adequate, the statistical analys is accurate and the results are well exposed.

The are two negative comment I have to say.

One is about originality of the study.

Ibanez-Sanz et al. reported other four studies with the same topic of their work, 3 of them with a large number of cases considered. Even with some differences in the methods, Cha JM et al. (AM J Gastroenterol, 2018) Doubeni CA et al. (J Am Board Fam Med, 2016) and Niedermaier T et al. (Clin Epidemiol 2018) get the same conclusions: the warmer months are at risk of reduced FIT-based screening performance.

On this regard, in the paper of Ibanez-Sanz et al. there are in some discrepances between results and conclusions compared to the discussion.

Their results and conclusions are that the main goal of the screening program (: the detection of advanced adenomas and interval cancers) is manteined during the warmer months of the year. On the contrary, in the discussione they write:

"One of the issues that emerge from our and previous findings is to implement effective measures to overcome the suboptimal performance of FIT during high ambient temperatures.

"Screening programs must implement circuits so that temperature does not impact the detection of advanced neoplasia"

"we believe that the more feasible to implement would be to instruct subjects in order to guarantee the cold chain after sample collection, decrease the number of FITs performed during warm months and decreasing the cut-off during warm periods".

Based on this discussion,  it appears that Authors believe more in the results of CHa JM, Doubeni CA and Niedermaier T , rather than in their conclusions.

I think Ibanez-Sanz and Coll. have to explain better why they believe in a remodulation of the screening programs in order to improve their efficacy during summer, while theresults of their work give an opposite conclusion.

Author Response

Reviewer #2:

This is a well written paper focused on the Fecal immunochemical based test (FIT) as screening tool for colorectal cancer (CRC). The Authors investigated if, in a large number of cases, temperature and humidity could affect the accuracy of the test. The final question the Authors ask themselves is if the screening organization has to spend any effort to maintain the fecal sample in a range of temperature and humidity in order not to decrease the screening program efficacy. The results of the work of Ibanez-Sanz et al. are that during summer there is a lower rate of positive FIT but not a lower adenomas and interval carcinomas detection rate. The methods are adequate, the statistical analyses is accurate and the results are well exposed. The are two negative comment I have to say:

One is about originality of the study. Ibanez-Sanz et al. reported other four studies with the same topic of their work, 3 of them with a large number of cases considered. Even with some differences in the methods, Cha JM et al. (AM J Gastroenterol, 2018) Doubeni CA et al. (J Am Board Fam Med, 2016) and Niedermaier T et al. (Clin Epidemiol 2018) get the same conclusions: the warmer months are at risk of reduced FIT-based screening performance. On this regard, in the paper of Ibanez-Sanz et al. there are in some discrepancies between results and conclusions compared to the discussion. Their results and conclusions are that the main goal of the screening program (the detection of advanced adenomas and interval cancers) is maintained during the warmer months of the year. On the contrary, in the discussion they write: "One of the issues that emerge from our and previous findings is to implement effective measures to overcome the suboptimal performance of FIT during high ambient temperatures. Screening programs must implement circuits so that temperature does not impact the detection of advanced neoplasia" "we believe that the more feasible to implement would be to instruct subjects in order to guarantee the cold chain after sample collection, decrease the number of FITs performed during warm months and decreasing the cut-off during warm periods". Based on this discussion, it appears that Authors believe more in the results of CHa JM, Doubeni CA and Niedermaier T, rather than in their conclusions. I think Ibanez-Sanz and Coll. have to explain better why they believe in a remodulation of the screening programs in order to improve their efficacy during summer, while the results of their work give an opposite conclusion.

We agree with the reviewer. We have rewritten part of the discussion to clarify this point and explain our conclusions.

Reviewer 3 Report

The authors reported an interesting study regarding temperature & humidity on FIT. The overall presentations are good. I lean to accepting the manuscript when the comments are well addressed.

My comments are listed as follows

-The M&M part can be broken into smaller sections with headings so that the presentation could be improved.

-The effect of temperature & humidity on FIT could be cumulative. Thus, the time factor should be provided. It was read that the sample was stored in refrigerators, what is the temperature in refrigerators? How long were the specimens stored? During transport from home to pharmacy, the sample, was exposed to ambient temperatures. When the specimens arrived at pharmacy, they got into the refrigerators again. It is kind of a freeze-thawing-freeze process. How long does the process take would be an important factor for investigating the study issue.

-My major concern is that not ambient temperature but other pre-analytical factors are actually the contributing factors for FIT. The freeze-thawing-freeze process prior to FIT obviously cause a much heavier change of temperature than the effect of different seasons. The potential effect of the freeze-thawing-freeze process should be addressed first.

-Other pre-analytical factors that potentially cause different positive rate of FIT is people's willing to get screen. Is the willing different in different season? It is a crucial factor needs control.

-Ambient humidity is generally a poorly controlled factor. Ambient humidity fluctuated heavily from space to space. In addition, some machines like air-condition or dehuminator do strong impact on ambient humidity. Investigating the effect of ambient humidity without the detailed information is too rough to be valuable. Biased conclusion is possible.

-In doing the logistic regression, both temperature and humidity can be input as numeric variable but not categorical variable. The data transformation is man-made and may cause bias or information missing in this step.

Author Response

The authors reported an interesting study regarding temperature & humidity on FIT. The overall presentations are good. I lean to accepting the manuscript when the comments are well addressed. My comments are listed as follows:

a) The M&M part can be broken into smaller sections with headings so that the presentation could be improved.

Done!

b) The effect of temperature & humidity on FIT could be cumulative. Thus, the time factor should be provided. It was read that the sample was stored in refrigerators, what is the temperature in refrigerators? How long were the specimens stored? During transport from home to pharmacy, the sample, was exposed to ambient temperatures. When the specimens arrived at pharmacy, they got into the refrigerators again. It is kind of a freeze-thawing-freeze process. How long does the process take would be an important factor for investigating the study issue.

We have included that: “Among participants who completed the FIT, the median time of return of the kit was 3 days”. Moreover, we have included a supplementary table (Table S1) that shows the temperature of conservation, maximum days of each step of the process and if the temperature is controlled.

c) My major concern is that not ambient temperature but other pre-analytical factors are actually the contributing factors for FIT. The freeze-thawing-freeze process prior to FIT obviously cause a much heavier change of temperature than the effect of different seasons. The potential effect of the freeze-thawing-freeze process should be addressed first.

Thank you for pointing this out. Though we agree with the reviewer, we cannot estimate the impact of the effect of the freeze-thawing process. We have included it as a limitation of the study.

d) Other pre-analytical factors that potentially cause different positive rate of FIT is people's willing to get screen. Is the willing different in different season? It is a crucial factor needs control.

We show the participation rate according to every quarter of year. We have added the sentence “The participation rate was similar between the 3rd quarter (July-September) and the 1st quarter (January-March)” but we have not included this table in the manuscript.

Participation

Quarter year of FIT performance

No

Yes

Total

1st quarter (January-March)

n

37,173

22,475

59,648

%

62.3

37.7

100.0

2nd quarter (April-June)

n

23,629

16,442

40,071

%

59.0

41.0

100.0

3rd quarter (July-September)

n

17210.0

9629.0

26839.0

%

64.1

35.9

100.0

4th quarter (October-December)

n

87742.0

47945.0

135687.0

%

64.7

35.3

100.0

Total

n

165754.0

96491.0

262245.0

%

63.2

36.8

100.0

e) Ambient humidity is generally a poorly controlled factor. Ambient humidity fluctuated heavily from space to space. In addition, some machines like air-condition or dehuminator do strong impact on ambient humidity. Investigating the effect of ambient humidity without the detailed information is too rough to be valuable. Biased conclusion is possible.

We agree with reviewer. It is an ecologic study and we cannot rule out this bias. We have included it as a limitation of the study.

f) In doing the logistic regression, both temperature and humidity can be input as numeric variable but not categorical variable. The data transformation is man-made and may cause bias or information missing in this step.

We have repeated the multivariate analysis including both temperature and humidity as continuous variables obtaining similar results. As they are very similar, we have not included the table in the manuscript but we have included in the manuscript that we have performed this analysis obtaining similar results. The results are:

Logistic regression of the probability of positive screening tests by temperature and humidity (95% CI).

OR2

95% CI

P-value

Sex

  Female

1.00

<0.001

  Male

1.67

1.58-1.77

Age1 (years)

1.03

1.03-1.04

<0.001

Screening episode

  Initial

1.00

<0.001

  Successive

0.68

0.64-0.73

Socioeconomic score1

1.01

1.00-1.01

<0.001

Maximum ambient temperature1

0.99

0.98-0.99

0.0003

Maximum ambient humidity1

1.00

1.00-1.01

0.14

1Continuous variable; 2All variables shown are included in the multivariate analysis. FIT: faecal immunochemical test.

Round 2

Reviewer 3 Report

I understand that the authors may not able to address my concerns directly. In the revision, they have highlighted the possible limitations and the biases in the discussion section. It is acceptable given the current dataset. However, I suggest to briefing the limitations also in the abstract so that the readers can understand the limitations and avoid biased inception at the very first in reading the paper. 

Author Response

We have included in the abstract that this is an ecological study.